# Isolation and Characterization of Chlorpyrifos-Degrading Gut Bacteria from Field-Collected Larvae of *Spodoptera frugiperda* (J.E. Smith) (Lepidoptera: Noctuidae)

**DOI:** 10.3390/biology14111468

**Published:** 2025-10-22

**Authors:** Ayatollah S. El-Zayat, Marwa N. Ahmed, Manar Sofy, Dalia E. El-Hefny, Nawal A. Alfuhaid, Dina El-Sayed, Hayam M. Fathy, Mona Awad

**Affiliations:** 1Department of Microbiology, Faculty of Agriculture, Cairo University, Giza 12613, Egypt; ayatollah.elzayat@agr.cu.edu.eg (A.S.E.-Z.); dina.gamal@agr.cu.edu.eg (D.E.-S.); hayam.fathy@cu.edu.eg (H.M.F.); 2School of Biotechnology, Nile University, Giza 12677, Egypt; marwa.nabil@agr.cu.edu.eg; 3Department of Economic Entomology and Pesticides, Faculty of Agriculture, Cairo University, Giza 12613, Egypt; m.kasem.2557@gmail.com; 4Pesticide Residues and Environmental Pollution Department, Central Agricultural Pesticide Laboratory, Agricultural Research Center, Dokki, Giza 12618, Egypt; daliaelhafny@yahoo.com; 5Department of Biology, College of Science and Humanities, Prince Sattam Bin Abdulziz University, Al-Kharj 11942, Saudi Arabia; n.alfuhaid@psau.edu.sa

**Keywords:** fall armyworm, gut microbiota, biodegradation, detoxification enzyme

## Abstract

The fall armyworm, *Spodoptera frugiperda* (J.E. Smith) (Lepidoptera: Noctuidae), is a highly invasive and economically important pest that inflicts significant damage on a wide range of host crops, particularly maize. One of the most challenging aspects of managing *S. frugiperda* populations is their capacity to develop resistance rapidly to various classes of chemical insecticides. Symbiotic microbes play vital roles in insect physiology and behavior, contributing to fundamental processes such as food digestion, nutrient acquisition, immune defense against pathogens, and even mating preferences, as well as insecticide resistance. In this study, four bacterial isolates capable of degrading chlorpyrifos were isolated from the gut of a field-collected population of *S. frugiperda*. These findings substantiate the concept of “detoxifying symbiosis,” wherein insect gut microbiota strengthen host survival in toxic environments. Harnessing such pesticide-degrading symbionts could pioneer innovative, microbe-based pest-management strategies that minimize reliance on synthetic chemicals.

## 1. Introduction

The fall armyworm (FAW), *Spodoptera frugiperda* (J.E. Smith) (Lepidoptera: Noctuidae), is a polyphagous pest native to tropical and subtropical Americas. The Food and Agriculture Organization (FAO) classified FAW as a global quarantine pest in 2016. Maize is the preferred host for FAW among infested countries, with yield losses of maize crops reaching up to 70% [1]. For a long time, chemical pesticides have been the main tactic for the management of FAWs [2]. Among them, chlorpyrifos is a broad-spectrum organophosphate insecticide commonly used against lepidopteran pests, which induces neurotoxicity and ultimately neuronal death [3]. The FAW is a pest with a high capacity to develop resistance to pesticides, as evidenced by the wide range of cases worldwide [2]. When exposed to pesticides, resistant insect pests enhance their tolerance through multiple mechanisms, including overexpression of pesticide detoxification and degradation enzymes [4,5] mutation of pesticide receptors [4,6], and altered cuticle thickness and composition [7]. Apart from these host-driven mechanisms, the gut microbiota of insects may also play an important role in alleviating the adverse effects of different insecticidal compounds [8].

More broadly, the diversity, abundance, and evolutionary success of insects are to some extent related to their relationships with symbiotic bacteria, which directly affect the performance of insects in nature [9]. Recent studies have revealed that the insect microbiome shapes their response to pesticides, either through direct degradation of insecticides by microbes that influence the host immune system [10] or through microbe-mediated activation or enhancement of insecticide toxicity [11,12]. Accompanying roles in nutrient provisioning and immune system stimulation, some microbial symbionts can break down complex molecules such as insecticides and promote insecticide resistance [10,13,14,15]. It is also notable that pathogenic bacteria can reside in host guts, only initiating or facilitating pathogenesis under certain conditions [16,17].

Insecticide-degrading bacteria are ubiquitous in nature and have been found in several orders of insects [18], such as Coleoptera [19,20], Diptera [21], Hemiptera [15], and Lepidoptera [22,23]. In Lepidoptera, in vitro assays demonstrated the potential of bacteria isolated from the gut of *Plutella xylostella* (Linnaeus) (Lepidoptera: Plutellidae) [23] and from resistant strains of *S. frugiperda* [22] to degrade several insecticides, including lambda-cyhalothrin, deltamethrin, chlorpyrifos ethyl, spinosad, and lufenuron. Moreover, insecticide-guided selection of strains of *S. frugiperda* led to the selection of insecticide-degrading bacteria, which were not present in the microbiota of susceptible, unselected larvae [22]. Similarly, Siddiqui et al. (2025) reported the isolation of several culturable bacterial species, including *Klebsiella variecola*, from *S. frugiperda* larvae treated with profluanilide, spinosad, and indoxacarb, further supporting the adaptive role of gut microbiota in pesticide detoxification [24].

To better understand this symbiotic interaction, we used the destructive invasive pest *S. frugiperda* to evaluate the effects of pesticide exposure on the diversity and ability of the gut bacterial community to metabolize pesticides. This study aimed to identify bacterial strains associated with pesticide resistance and uncover potential mechanisms. We hypothesized that gut bacteria would enhance the host’s ability to metabolize pesticides. To address this principle, the following methods were hypothesized: (i) isolate and identify the chlorpyrifos-degrading gut bacteria from the field collection of the invasive pest *S. frugiperda* larvae, (ii) evaluate chlorpyrifos biodegradation ability through an in vitro assay, and (iii) assess the impact of particular bacterial taxa that have the ability to degrade chlorpyrifos directly in the gut.

## 2. Materials and Methods

### 2.1. Sampling

A total of 50 *S. frugiperda* larvae were obtained from a maize field in Giza Governorate located at the following coordinates: “29°42′49.4″ N 31°17′12.4″ E” during summer 2022. In this field, insecticides are used periodically for the management of *S. frugiperda*. The collected larvae were transported directly to the laboratory in a cold container. The larvae were starved for 24 h before dissection.

### 2.2. Pesticides

Commercial-grade insecticide chlorpyrifos (chlorpyrifos^®^ 50% EC, Dow Agro Sciences, Manchester, UK), pyrethroid insecticide lambda-cyhalothrin (Axone^®^ 5% EC, 5 ppm; Jiangsu Changqing Agrochemical, Yangzhou, China), the diamide insecticide chlorantraniliprole (Coragen 20% SC, 0.6 ppm; Coragen^®^ 20% SC, DuPont, Nemours, France), and the carbamate insecticide Lannate (Lannit^®^ 90% WP, 0.5 ppm; Rotam Agrochemical, Zone Kunshan, Jiangsu, China) were used in the current study. All the tested insecticides are diluted in distilled water.

### 2.3. Isolation of Chlorpyrifos-Degrading Gut Bacteria

All larvae were surface-disinfected by immersion in 70% ethanol for 2 min, followed by two rinses with sterile distilled water. Under aseptic conditions and using a stereomicroscope (Bergamo, Italy), intact guts of ten *S. frugiperda* larvae were dissected and pooled into a sterile 15 mL Falcon tube containing 9 mL of sterile M9 salt solution [25]. To isolate chlorpyrifos-degrading gut bacteria, an enrichment culture technique was employed. One milliliter of the pooled gut homogenate was inoculated into M9 mineral medium (MM9) at a final concentration of 20% (*v*/*v*). The MM9 medium was prepared using the following components: 100 mL of 10× M9 salts (Na_2_HPO_4_·2H_2_O, 75.2 g/L; KH_2_PO_4_, 30 g/L; NaCl, 5 g/L; NH_4_Cl, 5 g/L), 1 mL of 1 M MgSO_4_·7H_2_O, and 1 mL of 1 M CaCl_2_, supplemented with chlorpyrifos (20% EC) at a concentration of 240 ppm as the sole carbon source.

The pH of the medium was adjusted to 7.0 ± 0.2. Cultures were incubated at 30 °C on a rotary shaker at 160 rpm for 14 days. Enrichment samples were collected on days 5, 7, 10, and 14. Aliquots (1 mL) were serially diluted and plated onto an MM9 agar medium containing chlorpyrifos (240 ppm) using the pour plate technique. Plates were incubated at 30 °C for 7 days. Emerging bacterial colonies were purified by repeated subculturing and preserved in 20% glycerol stocks at −80 °C for further identification and biodegradation analysis.

### 2.4. Molecular Identification of Bacterial Isolates

The purified isolates were grown on nutrient broth at 30 °C for 48 h. Total genomic DNA from the selected strains was extracted and purified using the Thermo Scientific GeneJET Kit, Waltham, MA, USA (K0721, K0722) according to the manufacturer’s protocol. 16S rRNA gene fragments were amplified using the Thermo Scientific Dream Taq Green PCR MasterMix (2×) kit according to the supplier’s protocol. PCR was carried out using forward (5′ CCAGCAGCCGCGGTAATACG 3′) and reverse (5′ATCGG(C/T)TACCTTGTTACGACTTC 3′) 16S rRNA primers [26,27]. 16S rRNA PCR amplicons were purified using a QIAquick PCR Purification Kit according to the supplier’s instructions (QIAGEN, Hilden, Germany). DNA sequencing was performed by the Sanger method [28] and performed by Macrogen (Seoul, Korea). The BLASTN program was used to submit the nucleotide sequences at the NCBI BLAST server (https://blast.ncbi.nlm.nih.gov/Blast.cgi, accessed on 28 March 2024). The partial 16S rDNA sequences (ranging from 1096 to 1328 bp and the average read quality (Q ≥ 30)) were deposited in NCBI GenBank under accession numbers PP504878-PP504881. The phylogenetic tree (neighbor-joining with bootstrap method) was constructed using MEGA11 [29]. The analysis featured 28 sequences, which correspond to the nearest matches found in the GenBank database.

### 2.5. Factors Affecting the Growth of Chlorpyrifos-Degrading Isolates

To optimize the growth of chlorpyrifos-degrading bacterial isolates, we assessed the effect of incubation temperature, pH, and incubation time. The flasks were loaded in triplicate with 20% of sterile MM9 broth media supplemented with 250 ppm chlorpyrifos. Each set of triplicate flasks was inoculated with 2% of fresh bacterial cultures (approx. 10^6^ CFU/mL). Each optimized parameter will be passed to the following one [25].

The effect of temperature was investigated by incubation at different temperatures (20, 25, 30, 35, and 40 °C) at pH 7 and 160 rpm for 5 days on a rotary shaker (VISION SCIENTIFIC CO., LTD., Model: VS-8480, Daejeon-Si, Korea). While studying the effect of pH, it was carried out with different pH ranges (4–9) and then incubated at the optimum temperature for each isolate and 160 rpm for 5 days on a rotary shaker. The optimum incubation time, which is required for maximum chlorpyrifos-degrading isolate growth, was determined at 0, 1, 2, 3, 4, 5, 6, 7, and 8 days. Followed by incubation at optimum temperature and pH for each isolate and 160 rpm on a rotary shaker. For growth study analysis of all factors, an aliquot of 1 mL of culture was withdrawn daily, and growth was evaluated as viable cell counts on nutrient agar media.

### 2.6. In Vitro Chlorpyrifos Biodegradation Assay

Four bacterial isolates—*Klebsiella quasipneumoniae* strain 60D, *Klebsiella pneumoniae* strain 64D, *Klebsiella pneumoniae* strain 66D, and *Klebsiella pneumoniae* strain 71D (approximately 10^6^ CFU/mL)—were used to inoculate MM9 broth medium supplemented with chlorpyrifos (250 ppm). Cultures were incubated at the optimal temperature for each isolate on a rotary shaker at 160 rpm for 5 days. Samples were collected at 1-day intervals. The colony-forming units (CFU/mL) of the incubated cultures were determined using the dilution plate counting method.

For pesticide biodegradation analysis, a modified QuEChERS extraction protocol was applied [30]. For sample preparation, 10 mL from each culture (in triplicate) was transferred into a 50 mL centrifuge tube, followed by the addition of 10 mL acetonitrile. Tubes were sealed and placed in a freezer at −18 °C for 15 min. Subsequently, 4 g of MgSO_4_ and 1 g of NaCl were added. Tubes were hand-shaken vigorously for 1 min, then centrifuged at 4000 rpm for 5 min at 5 °C. One milliliter of the supernatant was transferred to a sealed GC vial for analysis. Residual chlorpyrifos in the media was quantified by GC–MS analysis using an HP 6890 gas chromatograph equipped HPLC system (Agilent HPLC 1260 infinite series; Agilent Technologies, Santa Clara, CA, USA) with an HP 7673 autosampler. The external standard method was employed to determine the insecticide content in each sample, and the degradation rate was calculated using the following formula:*ρ* = (1 − *A*_1_/*A*_0_) × 100%(1)
where *A*_1_ represents the insecticide content in the MM9 medium after bacterial inoculation, and *A*_0_ represents the insecticide content in the uninoculated control [31]. Each sample was analyzed in three technical replicates, and its mean result was considered one biological replicate.

### 2.7. Screening of Biodegradation Capabilities by Chlorpyrifos-Degrading Isolates to Other Pesticides

To evaluate the broader biodegradation potential of the chlorpyrifos-degrading isolates, each isolate was streaked in triplicate onto MM9 agar plates supplemented with one of three different pesticide groups, each serving as the sole carbon source. The tested pesticides included the pyrethroid lambda-cyhalothrin (Axone 5% EC, 5 ppm), the diamide chlorantraniliprole (Coragen 20% SC, 0.6 ppm), and the carbamate Lannate (Lannate 90% WP, 0.5 ppm), each applied at its recommended field concentration. A negative control (uninoculated plates) was also included to ensure that no growth occurred in the absence of bacterial inoculation. The inoculated plates were incubated at 30 °C for 7 days. Bacterial growth on these selective media was considered indicative of the isolate’s potential biodegradation capability for the respective pesticide. This screening served as a preliminary assessment of the metabolic versatility of the isolates in degrading structurally and functionally distinct classes of insecticides.

### 2.8. In Vivo Chlorpyrifos Biodegradation Assay

#### 2.8.1. Experimental Design

A laboratory strain of *S. frugiperda* was used to determine the biodegradability of chlorpyrifos-degrading isolates in vivo. Egg clusters of *S. frugiperda* were used in two main groups, as demonstrated in Figure 1. First group (mono-associated): The egg clusters were immersed in 70% ethanol for 45 s and rinsed again in autoclaved distilled water three times. Disinfected eggs were maintained in a 90 mm plastic Petri dish and allowed to dry on a clean bench at room temperature. After hatching, larvae were reared on castor leaves (*Ricinus communis* L.) dipped in an antibiotic mixture containing Penicillin G, Polymyxin B, and erythromycin dissolved in sterile water at a concentration of 150, 150, and 600 μg/mL, respectively, to disrupt the *S. frugiperda* gut microbiome following protocols in [12,14]; leaves were replaced every 24 h, and antibiotic treatment was applied until the larvae molted to the third instar. The third instar larvae were reared individually in a clean, sterile plastic cup. The larvae were orally inoculated individually by daily feeding with castor leaf disks (5 cm) that had been dipped for 45 s in suspensions (approx. 10^6^ CFU/mL) of the chlorpyrifos-degrading isolates *Klebsiella quasipneumoniae* strain 60D, *Klebsiella pneumoniae* strain 64D, *Klebsiella pneumoniae* strain 66D, and *Klebsiella pneumoniae* strain 71D and, subsequently, air-dried under aseptic conditions at room temperature. The second group (control groups): Egg clusters were divided into two subgroups: (i) egg clusters that were disinfected as mentioned above and treated with an antibiotic mixture until they reached the 3rd larval instar, and (ii) egg clusters that were not treated with antibiotics, and the larvae were reared on castor leaves without any additional treatment. The third larval instar of the control groups was maintained daily with castor leaf disks (5 cm) dipped in distilled water until it reached the 4th larval instar. To validate the effect of antibiotics on the gut bacteria, larval weight and gut bacterial count were estimated. The thirty-third larval instar of *S. frugiperda* in all the above-mentioned treatments was weighed, and then the thirty larvae were freshly dissected in 10 mL of sterilized M9 salts, and bacterial counting was applied on both nutrient agar and MM9 agar media containing chlorpyrifos (250 ppm).

#### 2.8.2. Bioassay of *S. frugiperda* 4th Instar Larvae

Chlorpyrifos toxicity bioassays on newly ecdysed 4th instar larvae of *S. frugiperda* were performed using 24 ppm. The leaf-dipping technique was used according to [32] as follows: For experiments, castor leaves were dipped in chlorpyrifos suspension (24 ppm) for 20 s and then allowed to dry in the air. Each treatment had five replicates; each replicate consisted of 50 larvae (*n* = 250). Leaves dipped in water were used in the control group. The larvae were allowed to feed on treated leaves for 24 h, and the surviving larvae were then transferred to a clean, dry container containing fresh, untreated leaves [33]. Larval mortality was recorded daily for five consecutive days after post-treatment was calculated. In addition, detoxification enzyme activity, pesticide residue in larval feces, and bacterial count in larval feces were investigated.

#### 2.8.3. Biochemical Assays

##### Insect Treatment and Sample Preparation

Fourth instar larvae of *Spodoptera frugiperda* were exposed to a chlorpyrifos suspension (24 ppm) as described above. At 24, 72, and 120 h post-treatment, 50 mg of fresh body weight from surviving larvae were collected for enzymatic activity analysis. Each time point included five biological replicates. Larvae were homogenized in 0.1 M phosphate buffer, pH 7.8 for acetylcholinesterase (AChE) activity, and pH 6.5 for glutathione S-transferase (GST) activity. The homogenates were centrifuged at 7000 rpm for 15 min at 4 °C, and the resulting supernatants were used for enzyme activity and total protein measurements. Crude enzyme extracts were prepared separately for each enzyme assay. Total protein content in each replicate was determined using Coomassie Brilliant Blue dye, with bovine serum albumin as the standard, following Bradford’s method [34].

##### Acetylcholine Esterase (AChE) Assay

AChE activity was determined following the method of Ellman et al. 1961 [35]. The reaction mixture consisted of 100 µL of enzyme extract and 50 µL of 0.075 M acetylthiocholine iodide (ATChI) as the substrate. The reaction was initiated by adding 50 µL of 0.01 M dithiobisnitrobenzoic acid (DTNB). The absorbance was recorded at 412 nm at one-minute intervals over a 5-min period. Enzyme activity was expressed in units per mg of protein [36].

##### Glutathione S-Transferase (GST) Assay

GST activity was measured according to the method described by [37]. The reaction mixture contained 10 µL of enzyme extract, 25 µL of 30 mM 1-chloro-2,4-dinitrobenzene (CDNB) as the substrate, 25 µL of 50 mM reduced glutathione (GSH), and 940 µL of 50 mM potassium phosphate buffer (pH 6.5), bringing the total volume to 1 mL. The optical density was measured at 340 nm at one-minute intervals over 5 min. GST activity was calculated using the extinction coefficient (ε = 9.6 mM^−1^ cm^−1^) [38].

#### 2.8.4. Quantification of Chlorpyrifos Residues in Larval Feces

The extraction and analysis of chlorpyrifos residues from *S. frugiperda* feces were performed with slight modifications to the method described by [39]. Approximately 50 mg of feces were collected and transferred to a PTFE tube, to which 3 mL of acetonitrile was added. The samples were vortexed for 1 min, left to stand for 15 min, and then centrifuged at 3500 rpm for 15 min. The supernatants were frozen at −20 °C for 1 h to precipitate fats, then filtered through 0.22 μm PTFE syringe filters. For solid-phase extraction (SPE) cleanup, each sample was mixed with 500 mg of MgSO_4_ and 75 mg of PSA (primary secondary amine). The mixture was vortexed for 60 s and centrifuged again at 3500 rpm for 5 min. Approximately 1 mL of the supernatant was filtered through a 0.22 μm PTFE filter in preparation for GC-ECD analysis.

In Schwantes et al., 2020, gas chromatography was used for chlorpyrifos residue quantification with some modifications [40], a gas chromatograph (HP 6890) equipped with an autosampler (HP 7673) and a dual micro-electron capture detector (μECD), using a capillary column (30 m × 0.32 mm × 0.25 μm, HP-5, 5% phenylmethyl polysiloxane). The injector and detector temperatures were set at 250 °C and 300 °C, respectively, with nitrogen as the carrier gas at a flow rate of 1.5 mL/min. The GC oven temperature program was set as follows: initial temperature at 160 °C held for 2 min, then increased at 10 °C/min to 200 °C and held for 5 min. An amount of 1 µL aliquot of the prepared sample was injected, and the retention time for chlorpyrifos was 6.5 min.

#### 2.8.5. Bacterial Count in Larval Feces

To ensure the definite isolates’ presence, 50 mg of feces were collected from each plastic cup. These feces samples were homogenized with 450 μL of sterilized PBS solution, and 100 μL of the homogenized feces were plated on MM9 media supplemented with chlorpyrifos and incubated for 5 days at 30 °C.

### 2.9. Data Analysis

Data was coded and entered using the statistical package SPSS V.22 and R (version 3.2.5). Data were tested for satisfying assumptions of parametric tests, and continuous variables were subjected to the Shapiro–Wilk and Kolmogorov–Smirnov tests for normality. Probability and percentile data were standardized for normality using Arcsine Square Root. Data were presented as mean and standard deviation. ANOVA followed by Tukey’s post hoc pairwise comparison was performed for all experimental groups to evaluate the recorded growth factors, degradation rate, enzymatic activities (AChE and GST), bacterial counting, and in vitro assays. Each group was analyzed using at least three replicates (five replicates for the in vivo assay). *p*-values < 0.05 were considered statistically significant. Data were visualized, when applicable, using R software (version 3.2.5).

## 3. Results

### 3.1. Isolation and Molecular Identification of Chlorpyrifos-Degrading Gut Bacteria

Four bacterial isolates were obtained and purified. Then, the NCBI BLAST software 2.17.0 was employed to sequence and identify the four strains, facilitating a comparison with analogous sequences found in the GenBank database. The BLAST analysis of the 16S rDNA sequences from strains 60D, 64D, 66D, and 71D exposed a 100% similarity to *Klebsiella quasipneumoniae* strain 60D, *Klebsiella pneumoniae* strain 64D, *Klebsiella pneumoniae* strain 66D, and *Klebsiella pneumoniae* strain 71D, respectively, as shown in Figure 2. The 16S rRNA sequences were deposited and assigned to GenBank, with accession numbers that span from PP504878 to PP504881. The phylogenetic tree, constructed through the UPGMA for the bacterial isolates, illustrated substantial phylogenetic relatedness (Figure 2).

### 3.2. Factors Affecting the Growth of Chlorpyrifos-Degrading Isolates

A series of experiments was carried out in batch cultures to determine the ideal culture conditions for obtaining the highest growth of four chlorpyrifos-degrading strains: *Klebsiella quasipneumoniae* 60D, *Klebsiella pneumoniae* 64D, *Klebsiella pneumoniae* 66D, and *Klebsiella pneumoniae* 71D.

#### 3.2.1. Effect of Incubation Temperature

All *Klebsiella* strains demonstrated temperature-dependent growth, with optimal ranges typically from 25 °C to 35 °C. The highest growth was significantly (*p* < 0.05) observed at 30 °C for strains *K. quasipneumoniae* 60D and *K. pneumoniae* 71D, while for strains *K. pneumoniae* 64D and *K. pneumoniae* 66D, the optimum temperature was 35 °C (Figure 3). Bacterial counts are expressed as CFU/mL on a logarithmic scale. The obtained data showed consistently high counts in *K. pneumoniae* 64D and *K. pneumoniae* 66D (approx. 9.1 Log_10_), followed by *K. quasipneumoniae* 60D (approx. 8.8 Log_10_), then *K. pneumoniae* 71D (approx. 8.3 Log_10_). All strains exhibited a significant decline in viability at 20 °C and 40 °C.

#### 3.2.2. Effect of pH

The influence of pH (ranging from 5 to 9) on the growth of *K. quasipneumoniae* 60D and *K. pneumoniae* strains 64D, 66D, and 71D was evaluated by measuring bacterial counts (CFU/mL, Log_10_). *K. quasipneumoniae* 60D exhibited a markedly enhanced growth at pH 9 (*p* < 0.05), reaching approximately 9 Log_10_, while maintaining significantly lower but relatively stable growth (approx. 5.4–5.6 Log_10_) across pH 5 to 8, suggesting a narrower pH optimum. *K. pneumoniae* 64D displayed a higher growth at a pH range from 7 to 9 with no significant difference between these pH ranges (approx. 6.9–7.1 Log_10_). In contrast, *K. pneumoniae* 66D demonstrated optimal growth at pH 8 (approx. 8.9 Log_10_), with reduced viability at both acidic and high alkaline conditions. *K. pneumoniae* 71D showed a uniform growth profile over the tested pH ranges (approx. 6.1–6.5 Log_10_), indicating a broad pH tolerance but without a distinct optimum. These findings highlight pH-dependent variability in growth responses among *Klebsiella* strains, with optimum growth at neutral to alkaline pH, and all strains, specifically *K. quasipneumoniae* 60D, were stimulated at a pH of 9, as shown in Figure 4.

#### 3.2.3. Effect of Incubation Time

The growth kinetics of four *Klebsiella* strains—*K. quasipneumoniae* 60D and *K. pneumoniae* strains 64D, 66D, and 71D—were monitored over an 8-day incubation period, with bacterial counts expressed as CFU/mL (Log_10_), Figure 5. All strains exhibited distinct growth patterns over time, with significant inter-strain differences. *K. quasipneumoniae* 60D exhibited a marked increase in growth, reaching a peak at day 5 (approx. 8.8 Log_10_), followed by a significant moderate decline through day 8 (*p* < 0.05). *K. pneumoniae* 64D demonstrated early and stable growth, with a high growth peak starting at day 1 (approx. 7.4 Log_10_), while cell counts plateaued with nominal oscillation from day 2 through day 8 (approx. 6.8–7.2 Log_10_). *K. pneumoniae* 66D showed sustained high-level growth from day 1 through day 6 (approx. 7–7.2 Log_10_), with minimal fluctuation, before peaking slightly at day 8 (approx. 7.7 Log_10_). Conversely, *K. pneumoniae* 71D exhibited a more gradual growth trajectory, achieving its highest count (approx. 8.2 Log_10_) on day 5, followed by a significant reduction in viability by day 8 (approx. 5.9 Log_10_), *p* < 0.05.

### 3.3. In Vitro Insecticide Biodegradability of Isolated Gut Bacteria

All four isolated bacterial strains exhibited strong chlorpyrifos-degrading capabilities. The highest degradation rate was observed after one day of inoculation, with *Klebsiella pneumoniae* strain 64D achieving 80.38%, followed by *K. quasipneumoniae* strain 60D (77.9%), *K. pneumoniae* strain 71D (65.52%), and *K. pneumoniae* strain 66D (51.3%) (Table 1). However, after five days of incubation, *K. pneumoniae* strain 66D showed the highest degradation efficiency at 61.14%. Nevertheless, it is not always possible to extrapolate biodegradation rates calculated for higher pesticide doses to lower concentrations [41], as strain 64D exhibited the lowest degradation rate at 25%.

### 3.4. Screening of Biodegradation Capabilities by Chlorpyrifos-Degrading Isolates to Other Pesticides

All four strains demonstrated the ability to degrade chlorantraniliprole (Coragen) and the carbamate insecticide (Lannate). Notably, only *K. quasipneumoniae* strain 60D exhibited the capacity to biodegrade the pyrethroid lambda-cyhalothrin (Table 2).

### 3.5. In Vivo Chlorpyrifos Biodegradation Assay

#### 3.5.1. Bacterial Counting in Larval Gut

The bacterial CFU/mL (Log10) of chlorpyrifos-degrading gut bacteria was determined under different conditions using NA (Figure 6A) and MM9 mineral media supplemented with chlorpyrifos (Figure 6B). A comparison of antibiotic-treated and non-antibiotic-treated larval groups revealed significant variations in bacterial counts (*p* < 0.05) grown on MM9 media supplemented with chlorpyrifos (approx. 2.2 Log_10_). However, the bacterial counts in antibiotic-treated and non-antibiotic-treated larvae groups on NA media did not differ significantly. These results suggest that MM9 mineral media, supplemented with chlorpyrifos as the sole carbon source, selectively enriched chlorpyrifos-degrading bacteria, particularly in non-antibiotic-treated larvae. Furthermore, while bacterial survival is similar across the groups on rich media, antibiotic treatment decreases bacterial colonization with chlorpyrifos under nutrient-limited settings.

#### 3.5.2. Larval Weight

The data presented in Table 3 indicate that *S. frugiperda* larvae treated with an antibiotic mixture for six days post-hatching exhibited a significant reduction in weight (8.57 mg) compared to the non-antibiotic-treated group (12.73 mg). This suggests that smaller body mass in antibiotic-treated larvae may result from the disruption of the gut microbiota, which adversely affects larval development and growth [10,22,42], or reduction in nutrient uptake [43].

#### 3.5.3. Mortality

The mortality rate of 4th instar *S. frugiperda* larvae after being exposed to chlorpyrifos suspension at 24 ppm exhibited varying levels of toxicity. As shown in Figure 7 and TS1, the Ab-treated larvae caused the maximum mortality rate (14%) after 24 h of treatment. The second-highest mortality rate was recorded with *K. pneumoniae* strain 71D (8%), followed by non-Ab-treated larva (5%), *K. quasipneumoniae* strain 60D (3.8%), *K. pneumoniae* strain 64D (1.6%), and *K. pneumoniae* strain 66D (0%) after 24 h of treatment. After 5 days of treatment, Ab-treated larvae exhibited higher larval mortality of 92%, followed by non-Ab-treated larvae (22%), *K. pneumoniae* strain 71D (18%), *K. pneumoniae* strain 64D (10%), *K. quasipneumoniae* strain 60D (8%), and *K. pneumoniae* strain 66D (4.8%).

#### 3.5.4. Detoxification Enzymes Activity

After being exposed to chlorpyrifos suspension (24 ppm), the AchE enzyme activity of *S. frugiperda* larvae was significantly reduced after 24 h compared with antibiotic-treated larvae. The mono-associated *K. quasipneumoniae* strain 60D shows the highest inhibition in AchE enzyme activity after being exposed to chlorpyrifos suspension by 3.5-fold compared to antibiotic-treated larvae. Followed by *K. pneumoniae* strain 64D, *K. pneumoniae* strain 71D, and *K. pneumoniae* strain 66D by about 2.96, 1.8, and 1.2-fold, respectively (Table 4). After three and five days from exposure to chlorpyrifos suspension, there is no significant difference in AchE enzyme activity of *S. frugiperda* larvae. The GST activity exhibited a different pattern; there was no significant difference in GST enzyme activity in *S. frugiperda* larvae after 1 and 3 days of exposure to chlorpyrifos suspension. A significant difference has been observed in the mono-associated *K. pneumoniae* strain 66D, *K. pneumoniae* strain 71D, *K. quasipneumoniae* strain 60D, and *K*. pneumoniae strain 64D by about 3.0-, 2.9-, 2.2-, and 1.8-fold inhibition, respectively, compared with antibiotic-treated larvae.

#### 3.5.5. Pesticide Residues in Feces (Larval Frass)

Residues of chlorpyrifos were detected in the feces of *S. frugiperda* larvae after 1, 3, and 5 days of exposure to chlorpyrifos suspension (Figure 8). On day 1, mono-associated larvae with *K. quasipneumoniae* strain 60D (approximately 119 μg/100 mg feces) and *K. pneumoniae* strain 64D (approximately 115 μg/100 μg feces) (*p* < 0.0001) showed the highest chlorpyrifos residues. By day 5, *K. quasipneumoniae* strain 60D still exhibited the highest fecal chlorpyrifos residues (approximately 61 μg/100 μg feces), followed by *K. pneumoniae* strain 66D (approximately 42 μg/100 μg feces) (*p* < 0.0001).

#### 3.5.6. Bacteria Counting in Larval Feces

The bacterial CFU counts (Log10) of chlorpyrifos-degrading gut bacteria in larval feces were assessed over time (Figure 9). These findings show that bacterial growth differed between antibiotic-treated larvae, non-antibiotic-treated larvae, and several isolated strains. Bacterial CFU counts in antibiotic-treated larvae were consistently low at all time points, with no development by day 5. Bacterial counts grew gradually in non-antibiotic-treated larvae, reaching around 6 log10 CFU/g by day 5. Individual bacterial isolates, including *Klebsiella* strains 60D, 66D, and 71D, followed similar growth trends, with CFU counts increasing from day 1 to day 5. Interestingly, strain 66 had the highest CFU counts, reaching over 7 log10 CFU/g by day 5, demonstrating its strong ability to degrade chlorpyrifos. Strain 64D showed a lower count by day 5 (5.2 log10 CFU/g). The results indicate that non-antibiotic-treated larvae include a resilient microbial population capable of efficiently degrading chlorpyrifos.

## 4. Discussion

*Spodoptera frugiperda*, as a key invasive pest, has the capacity to rapidly develop resistance to insecticides [44]. At the field level, the insect, alongside its microbial community, remains subject to constant selection pressures. An increase in the resistance of *S. frugiperda* to different insecticides has various underlying mechanisms, including elevated activity of the detoxifying enzymes, as well as insensitivity of the target [45]. In addition, as observed in other insects, resistance may also be associated with detoxification processes mediated by symbiotic microorganisms [15,19,21]. We analyzed the symbiotic gut bacteria in *S. frugiperda*, revealing their pivotal role in the biodegradation of pesticides. In this study, we successfully isolated four isolates of chlorpyrifos-degrading bacteria from a field-collected population of *S. frugiperda*. Our isolations identified isolates belonging to the phylum Pseudomonadota (formerly Proteobacteria), an abundant bacterial group in the insect gut that plays a significant role in insecticide degradation [13,24]. These isolates were identified using 16S rDNA sequencing as *Klebsiella quasipneumoniae* strain 60D (PP504878), *Klebsiella pneumoniae* strain 64D (PP504879), *Klebsiella pneumoniae* strain 66D (PP504880), and *Klebsiella pneumoniae* strain 71D (PP504881), as shown in (Figure 2). Notably, previous studies have demonstrated the chlorpyrifos-degrading potential of several bacterial species, including the genus *Klebsiella* [46,47,48].

When these gut bacteria were cultured with insecticides as the sole carbon source, we anticipated they would metabolize the insecticides ingested by their hosts, thereby contributing to insecticide tolerance of the host system [49]. To evaluate the degradation ability of the four chlorpyrifos-degrading gut bacterial isolates, the degradation rate was assessed by measuring the residual chlorpyrifos in MM9 medium using GC–MS analysis. Most of the isolates obtained from the midgut of *S. frugiperda* exhibited strong chlorpyrifos-degrading capabilities, as shown in Table 2. Our results indicated that *Klebsiella pneumoniae* strain 64D showed the highest degradation rate after 1 day (80.38%), but its efficiency declined to 25% after 5 days, whereas *K. pneumoniae* strain 66D maintained the highest activity at this stage (61.14%). The decline in strain 64D may reflect enzyme kinetic limitations or substrate/product inhibition, where reduced chlorpyrifos availability or accumulation of metabolites suppresses further enzymatic activity and bacterial growth, reflecting the complex interplay between enzyme induction, substrate concentration, and metabolic feedback regulation [50].

The importance of gut microbiota in insecticide degradation and resistance is becoming increasingly recognized. *Bacillus cereus* isolated from *Plutella xylostella* can degrade up to 20% of indoxacarb, highlighting its role in host growth and metabolism [23]. Moreover, *Enterococcus* sp. has been shown to enhance resistance to chlorpyrifos in *P. xylostella*, whereas proteobacteria, including *Enterobacter* sp. and *Serratia* sp., reduce it, suggesting complex and species-specific interactions between gut microbiota and insecticide resistance mechanisms [10]. With growing evidence demonstrating that gut microbiota confers insecticide tolerance and resistance across different arthropods [49,51]. Presumably, bacteria can influence this resistance through three non-exclusive mechanisms: (1) mitigating the fitness costs associated with insecticide resistance; (2) directly degrading insecticides; and (3) regulating the host’s detoxification metabolism [49,51,52]. To gain deeper insight into the interaction between gut bacteria and insecticide resistance, we examined the effect of gut bacterial communities in *S. frugiperda*. The results of the in vivo chlorpyrifos biodegradation assessment demonstrated that symbiotic gut bacteria play a critical role in enhancing host tolerance to chlorpyrifos. The mortality rate of 4th instar *S. frugiperda* larvae exposed to a 24 ppm chlorpyrifos suspension varied significantly, indicating differential toxicity levels. Notably, larvae mono-associated with *K. pneumoniae* strain 66D exhibited the greatest reduction in mortality by 19.16-fold compared to antibiotic-treated larvae. This was followed by larvae colonized with *K. quasipneumoniae* strain 60D (11-fold mortality reduction), *K. pneumoniae* strain 64D (9.2-fold mortality reduction), and *K. pneumoniae* strain 71D (5.11-fold mortality reduction) after five days of exposure. These findings indicate that specific isolates contribute to chlorpyrifos detoxification and enhance host survival [15,48]. Residues of chlorpyrifos were detected in the feces of *S. frugiperda* larvae after 1, 3, and 5 days of exposure to the chlorpyrifos suspension. The mono-associated larvae with *K. quasipneumoniae* strain 60D and *K. pneumoniae* strain 64D exhibited the highest chlorpyrifos residues in their feces after one day of exposure (Figure 6). This pattern is consistent with the observed mortality results, in which larvae colonized with *K. quasipneumoniae* strain 60D and *K. pneumoniae* strain 64D showed 11-fold and 9.2-fold reductions in mortality, respectively.

Further supporting a role for gut symbionts in pesticide detoxification, acetylcholinesterase (AChE) activity was significantly inhibited in larvae exposed to chlorpyrifos compared to antibiotic-treated controls. Mono-association with *K. quasipneumoniae* strain 60D resulted in the strongest AChE inhibition (3.5-fold), followed by strains 64D (2.96-fold), 71D (1.8-fold), and 66D (1.2-fold), as shown in Table 4. This inhibition is likely related to the chlorpyrifos mode of action, an organophosphate insecticide that irreversibly inhibits acetylcholinesterase (AChE), leading to the accumulation of acetylcholine at synaptic junctions [53]. These observations support those of Xiao et al., 2022, who reported that the gut symbiont *Stenotrophomonas maltophilia* can modulate host detoxification enzyme responses under chlorpyrifos exposure [54]. In contrast, GST activity exhibited a different pattern. No significant changes were observed in GST activity during the three days of chlorpyrifos exposure. However, by the fifth day, significant inhibition of GST activity was observed in larvae mono-associated with *K. pneumoniae* strain 66D, *K. pneumoniae* strain 71D, *K. quasipneumoniae* strain 60D, and *K. pneumoniae* strain 64D, with reductions of approximately 3.0-, 2.9-, 2.2-, and 1.8-fold, respectively, compared to antibiotic-treated larvae. Chlorpyrifos detoxification primarily occurs through conjugation with glutathione (GSH), a process catalyzed by glutathione S-transferase (GST) [55,56]. Moreover, chlorpyrifos can induce the generation of reactive oxygen species (ROS) that disrupt normal cellular development and differentiation [57,58]. These findings further support the role of gut microbiota in modulating host detoxification pathways, specifically through the regulation of AChE and GST activity [54]. Overall, this study extends current knowledge on pesticide–microbe interactions by demonstrating chlorpyrifos biodegradation in a different *Spodoptera* species and by identifying *Klebsiella* spp. among the key bacterial taxa involved, thereby providing novel perspectives on the microbial basis of pest resistance.

## 5. Conclusions

In summary, the isolation of *Klebsiella* strains from the gut of *Spodoptera frugiperda* and their remarkable capability to degrade chlorpyrifos both in vitro (up to 80.4% by strain 64D) and in vivo (notably, strain 66D afforded a 19.16-fold reduction in larval mortality relative to antibiotic-treated controls) underscores the pivotal role of gut-associated microbes in insecticide detoxification. These findings substantiate the concept of “detoxifying symbiosis,” wherein insect gut microbiota strengthen host survival in toxic environments. Supporting this, acetylcholinesterase (AChE) activity was significantly inhibited in larvae exposed to chlorpyrifos compared to antibiotic-treated control groups. Overall, harnessing such pesticide-degrading symbionts could pioneer innovative, microbe-based pest-management strategies that minimize reliance on synthetic chemicals.

## Figures and Tables

**Figure 1 biology-14-01468-f001:**
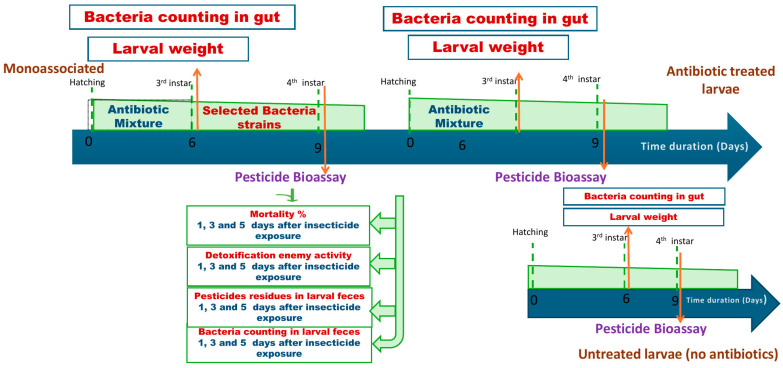
Experimental design of the in vivo chlorpyrifos biodegradation assay in *Spodoptera frugiperda* larvae.

**Figure 2 biology-14-01468-f002:**
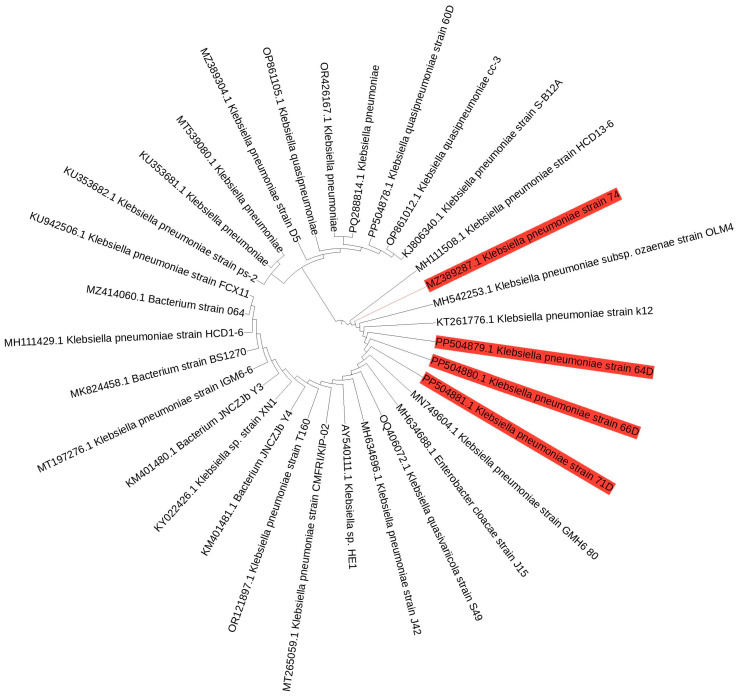
A neighbor-joining phylogenetic tree based on 16S rRNA gene sequences of four bacterial isolates (highlighted in red color) and their closest matches from the NCBI database.

**Figure 3 biology-14-01468-f003:**
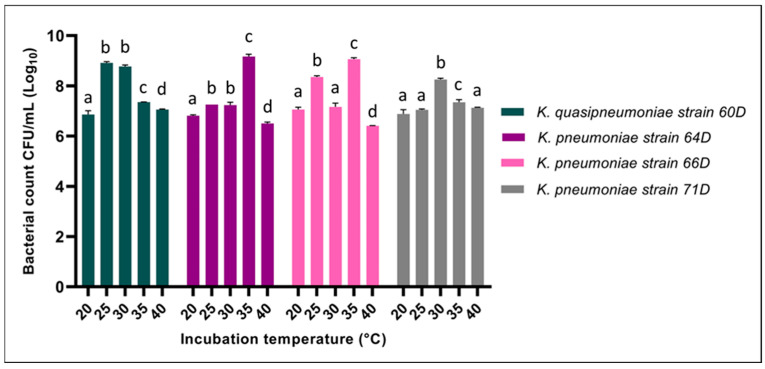
Effect of incubation temperature on the growth of *Klebsiella* strains. Bacterial counts (expressed as Log_10_ CFU/mL) of *K. quasipneumoniae* strain 60D, *K. pneumoniae* strain 64D, *K. pneumoniae* strain 66D, and *K. pneumoniae* strain 71D. Different letters denote statistically significant differences (*p* < 0.05).

**Figure 4 biology-14-01468-f004:**
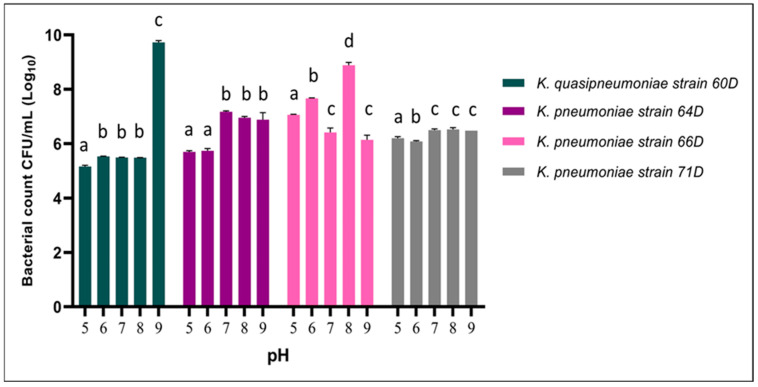
Effect of pH range on the growth of *Klebsiella* strains. Bacterial counts (expressed as Log_10_ CFU/mL) of *K. quasipneumoniae* strain 60D, *K. pneumoniae* strain 64D, *K. pneumoniae* strain 66D, and *K. pneumoniae* strain 71D. Different letters denote statistically significant differences (*p* < 0.05).

**Figure 5 biology-14-01468-f005:**
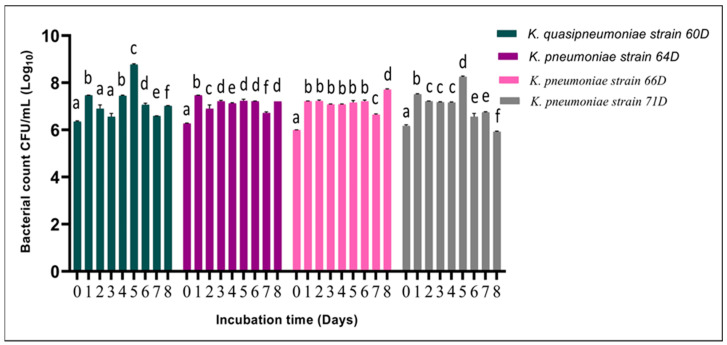
Effect of incubation time on the growth of *Klebsiella* strains. Bacterial counts (expressed as Log_10_ CFU/mL) of *K. quasipneumoniae* strain 60D, *K. pneumoniae* strain 64D, *K. pneumoniae* strain 66D, and *K. pneumoniae* strain 71D. Different letters denote statistically significant differences (*p* < 0.05).

**Figure 6 biology-14-01468-f006:**
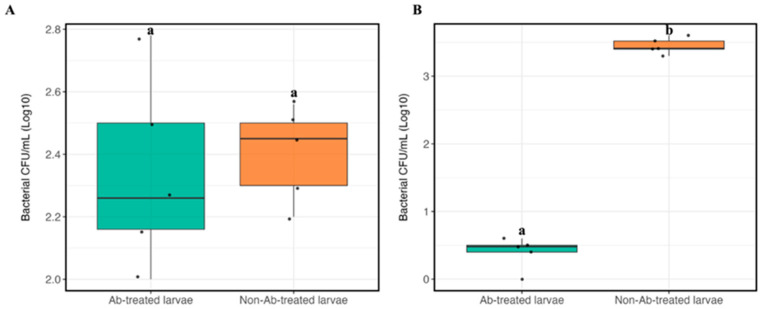
Bacterial load in Ab-treated and non-Ab-treated larvae grown on different media. (**A**) Bacterial load (log_10_ CFU/mL) on nutrient agar (NA), (**B**) bacterial load on minimal medium (MM9). Different letters denote statistically significant differences (*p* < 0.05).

**Figure 7 biology-14-01468-f007:**
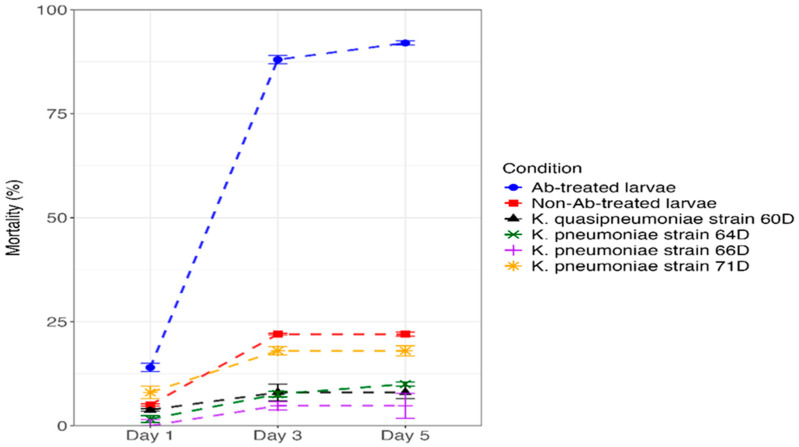
Mortality rates (%) of 4th *S. frugiperda* larvae after being exposed to chlorpyrifos.

**Figure 8 biology-14-01468-f008:**
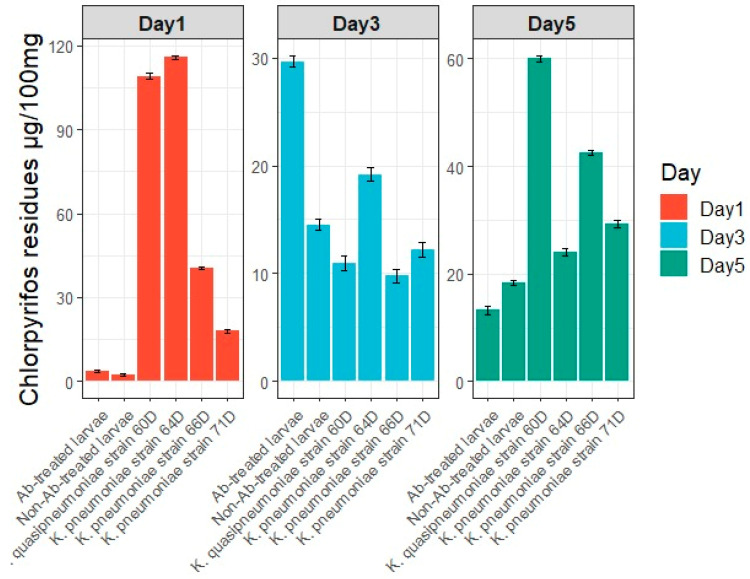
In vivo analysis of chlorpyrifos biodegradation by gut bacteria, as detected in the feces of *Spodoptera frugiperda* larvae after 1, 3, and 5 days of exposure to a chlorpyrifos suspension.

**Figure 9 biology-14-01468-f009:**
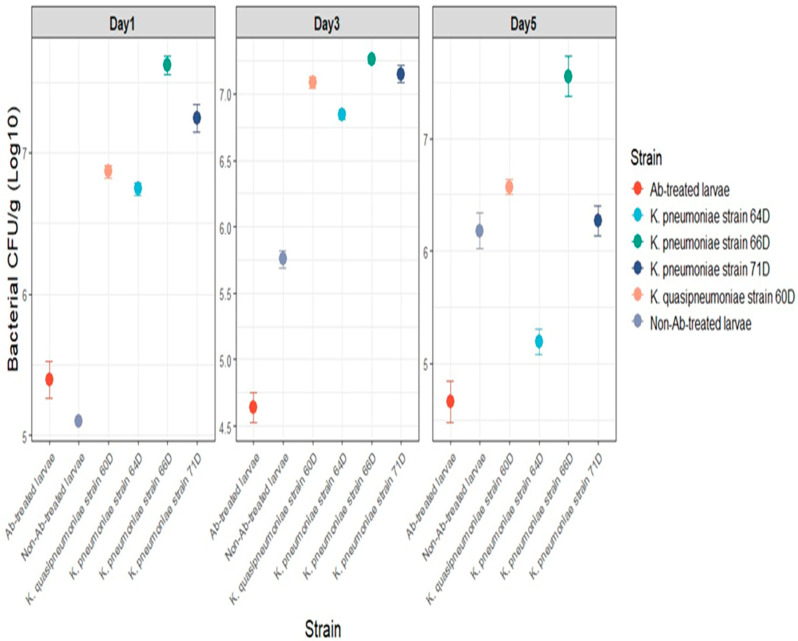
Bacterial CFU counts (Log_10_ CFU/g) of chlorpyrifos-degrading gut bacteria recovered from larval feces across three time points (day 1, day 3, and day 5).

**Table 1 biology-14-01468-t001:** Degradation rate % (±SD) of chlorpyrifos in MM9 media after 1, 3, and 5 days of bacteria inoculation (*n* = 3).

Groups	1 Day	3 Days	5 Days
*Klebsiella quasipneumoniae* strain 60D	77.9 ± 0.91 ^b^	45.13 ± 1.94 ^b^	57.94 ± 1.81 ^ab^
*Klebsiella pneumoniae* strain 64D	80.38 ± 0.68 ^a^	69.8 ± 0.32 ^a^	25 ± 1.82 ^c^
*Klebsiella pneumoniae* strain 66D	51.3 ± 0.15 ^d^	46.93 ± 1.64 ^b^	61.14 ± 2.12 ^a^
*Klebsiella pneumoniae* strain 71D	65.52 ± 0.74 ^c^	50.15 ± 2.06 ^b^	53.08 ± 1.82 ^b^

Means that those that do not share a letter are significantly different.

**Table 2 biology-14-01468-t002:** The biodegradation capabilities of chlorpyrifos-degrading isolates to other pesticides.

Strain/Pesticide	Chlorantraniliprole (Coragen) 0.6 ppm	Carbamate (Lannate) 0.5 ppm	Pyrethroid (Lambda-Cyhalothrin) 5 ppm
*K. quasipneumoniae* 60D	Positive	Positive	Positive
*K. pneumoniae* 64D	Positive	Positive	Negative
*K. pneumoniae* 66D	Positive	Positive	Negative
*K. pneumoniae* 71D	Positive	Positive	Negative

**Table 3 biology-14-01468-t003:** Larval weight (mg) of antibiotic-treated and non-antibiotic-treated application.

	Antibiotic-Treated (mg)	Non-Antibiotic-Treated (mg)	*t*-Test	*p*-Value
Larval weight (mg)	8.57 ± 1.89	12.73 ± 0.47	−3.02	0.039

**Table 4 biology-14-01468-t004:** Mean (±SD) of enzymatic activity of *S. frugiperda* after 1, 3, and 5 days of being exposed to chlorpyrifos.

Enzyme	Groups	1 Day	3 Days	5 Days
**AchE** **(Mmole/mg of protein)**	Ab-treated larva	0.74 ± 0.12 ^a^	0.61 ± 0.02 ^a^	0.43 ± 0.05 ^a^
	Non-Ab-treated larva	0.32 ± 0.05 ^cd^	0.43 ± 0.07 ^a^	0.36 ± 0.03 ^a^
	*Klebsiella quasipneumoniae* strain 60D	0.21 ± 0.06 ^d^	0.48 ± 0.06 ^a^	0.42 ± 0.01 ^a^
	*Klebsiella pneumoniae* strain 64D	0.25 ± 0.05 ^cd^	0.58 ± 0.08 ^a^	0.28 ± 0.08 ^a^
	*Klebsiella pneumoniae* strain 66D	0.58 ± 0.06 ^ab^	0.61 ± 0.03 ^a^	0.46 ± 0.09 ^a^
	*Klebsiella pneumoniae* strain 71D	0.41 ± 0.03 ^bcd^	0.66 ± 0.03 ^a^	0.26 ± 0.02 ^a^
	*P*-Value	0.001	0.171	0.020
	*F*-Value(df)	14.54(6)	1.80(6)	3.71(6)
**GST (Mmole/mg of protein)**	Ab-treated larva	0.44 ± 0.25 ^a^	0.38 ± 0.07 ^b^	0.87 ± 0.07 ^a^
	Non-Ab-treated larva	0.3 ± 0.05 ^a^	0.79 ± 0.04 ^a^	0.46 ± 0.03 ^b^
	*Klebsiella quasipneumoniae* strain 60D	0.2 ± 0.03 ^a^	0.24 ± 0.08 ^b^	0.39 ± 0.03 ^b^
	*Klebsiella pneumoniae* strain 64D	0.28 ± 0.01 ^a^	0.45 ± 0.09 ^b^	0.47 ± 0.16 ^b^
	*Klebsiella pneumoniae* strain 66D	0.41 ± 0.14 ^a^	0.35 ± 0.05 ^b^	0.29 ± 0.13 ^b^
	*Klebsiella pneumoniae* strain 71D	0.35 ± 0.07 ^a^	0.8 ± 0.12 ^a^	0.3 ± 0.11 ^b^
	*P*-Value	0.479	0.001	0.001
	*F*-Value(df)	0.97(6)	10.96(6)	8.94(6)

Means that do not share a letter are significantly different.

## Data Availability

All data from the study have been presented in the manuscript.

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
