# Peer review of "Isolation and Characterization of Chlorpyrifos-Degrading Gut Bacteria from Field-Collected Larvae of Spodoptera frugiperda (J.E. Smith) (Lepidoptera: Noctuidae)"

_biology, 2025, doi:10.3390/biology14111468_

Round 1
Reviewer 1 Report
Comments and Suggestions for Authors
The manuscript describes the isolation and characterization of chlorpyrifos-degrading gut bacteria from Spodoptera frugiperda and evaluates their role in insecticide tolerance both in vitro and in vivo. The study addresses an important and timely topic, as insecticide resistance in S. frugiperda represents a major agricultural challenge. The authors successfully combined microbiological, molecular, and bioassay approaches, and the manuscript provides valuable insights into the concept of “detoxifying symbiosis.” These are clear strengths of the work.
However, there are several methodological, analytical, and presentation issues that should be addressed before the manuscript can be considered for publication. Below, I provide detailed comments in numbered points to which the authors should respond.
- [Line 18–28, Simple Summary]
The Simple Summary is rather lengthy and repeats content from the Abstract. Please consider shortening it and emphasizing the key conclusions in a clearer way (the role of bacteria in chlorpyrifos detoxification and their potential in crop protection).
- [Lines 40–42]
Species names and strains – please use consistent nomenclature. In some places you write “Klebsiella quasipneumoniae strain 60D,” while in others only “K. quasipneumoniae 60D.” This should be standardized.
- [Lines 52–67, Introduction]
The introduction is quite long and contains many general details about insect resistance to pesticides. Consider shortening it and more clearly emphasizing the research gap (the role of gut bacteria in chlorpyrifos detoxification in S. frugiperda).
- [Lines 126–137, Molecular identification]
There is no information provided about the length of the obtained 16S rRNA sequences or the quality of the reads. For clarity, please include the length range (bp) and the average read quality.
- [Lines 138–141]
In the phylogenetic analysis, the UPGMA method was used. Currently, more robust approaches (e.g., Maximum Likelihood or Neighbor-Joining with bootstrap support) are preferred. Please consider reanalyzing the data using one of these methods or provide a clear justification for the choice of UPGMA.
- [Lines 142–157, Growth factors]
Detailed statistical information is missing (e.g., p-values, number of biological replicates). Please provide these details to clarify the analyses.
- [Lines 158–180, In vitro biodegradation]
Please specify whether the reported replicates are biological or technical. This issue recurs in multiple sections and should be clarified and standardized throughout the manuscript.
- [Lines 185–196, Screening for other pesticides]
The description of biodegradation of other pesticides is very brief—limited to “bacterial growth was observed.” Please consider providing quantitative results (e.g., decrease in pesticide concentration measured by HPLC/GC-MS).
- [Lines 290–300, Data analysis]
Only simple ANOVA tests are used. For repeated measurements (e.g., growth curves, enzyme activity over time), a mixed-effects model or RM-ANOVA would be more appropriate. Please consider revising the statistical approach.
- [Lines 301–312, Results – Identification]
Figure 2: the legend refers to “neighbor-joining,” while the Methods section specifies UPGMA. Please unify the terminology.
- [Lines 394–405, in vivo bacterial count]
The results are described only qualitatively (“significant differences,” “substantial variations”). Please provide the exact values and specify the statistical tests used.
- [Lines 486–565, Discussion]
The discussion is lengthy and at times repeats the results. Please consider shortening it and placing stronger emphasis on the novelty of the work (isolation of Klebsiella strains and demonstration of their impact on larval survival).
Author Response
Reviewer 1
The manuscript describes the isolation and characterization of chlorpyrifos-degrading gut bacteria from Spodoptera frugiperda and evaluates their role in insecticide tolerance both in vitro and in vivo. The study addresses an important and timely topic, as insecticide resistance in S. frugiperda represents a major agricultural challenge. The authors successfully combined microbiological, molecular, and bioassay approaches, and the manuscript provides valuable insights into the concept of “detoxifying symbiosis.” These are clear strengths of the work.
However, there are several methodological, analytical, and presentation issues that should be addressed before the manuscript can be considered for publication. Below, I provide detailed comments in numbered points to which the authors should respond.
- [Line 18–28, Simple Summary]
The Simple Summary is rather lengthy and repeats content from the Abstract. Please consider shortening it and emphasizing the key conclusions in a clearer way (the role of bacteria in chlorpyrifos detoxification and their potential in crop protection).
Response: Thank you for your valuable suggestion. The Simple Summary has been shortened and revised.
- [Lines 40–42]
Species names and strains – please use consistent nomenclature. In some places you write “Klebsiella quasipneumoniae strain 60D,” while in others only “K. quasipneumoniae 60D.” This should be standardized.
Response: We standardized all nomenclature to be Klebsiella in abstract section.
- [Lines 52–67, Introduction]
The introduction is quite long and contains many general details about insect resistance to pesticides. Consider shortening it and more clearly emphasizing the research gap (the role of gut bacteria in chlorpyrifos detoxification in S. frugiperda).
Response: Thank you for your insightful comment. The Introduction section has been revised and relevant references have also been added to support this context regarding the role of gut bacteria, particularly Klebsiella bacteria, in the degradation of chlorpyrifos.
- [Lines 126–137, Molecular identification]
There is no information provided about the length of the obtained 16S rRNA sequences or the quality of the reads. For clarity, please include the length range (bp) and the average read quality.
Response: The sequence length range (bP) and the average read quality were added.
- [Lines 138–141]
In the phylogenetic analysis, the UPGMA method was used. Currently, more robust approaches (e.g., Maximum Likelihood or Neighbor-Joining with bootstrap support) are preferred. Please consider reanalyzing the data using one of these methods or provide a clear justification for the choice of UPGMA.
Response: The used method was corrected as we used Neighbor-Joining with bootstrap method.
- [Lines 142–157, Growth factors]
Detailed statistical information is missing (e.g., p-values, number of biological replicates). Please provide these details to clarify the analyses.
Response: As stated in lines 145–146, the flasks used for the growth factor experiments were prepared in triplicate and filled with 20% sterile MM9 broth medium supplemented with 250 ppm chlorpyrifos. Each set of triplicate flasks was then inoculated with the corresponding isolated strain. p-value and the statistical method was described in data analysis section.
- [Lines 158–180, In vitro biodegradation]
Please specify whether the reported replicates are biological or technical. This issue recurs in multiple sections and should be clarified and standardized throughout the manuscript.
Response: Thank you for your observation. All replicates mentioned in this study are biological replicates.
- [Lines 185–196, Screening for other pesticides]
The description of biodegradation of other pesticides is very brief—limited to “bacterial growth was observed.” Please consider providing quantitative results (e.g., decrease in pesticide concentration measured by HPLC/GC-MS).
Response: This study represents a preliminary assessment and screening intended to determine whether our strains can grow in the presence of other pesticides. The results will guide future research to explore their potential for degrading additional pesticides and to quantify degradation rates at different pesticide concentrations.
- [Lines 290–300, Data analysis]
Only simple ANOVA tests are used. For repeated measurements (e.g., growth curves, enzyme activity over time), a mixed-effects model or RM-ANOVA would be more appropriate. Please consider revising the statistical approach.
Response: ANOVA followed by Tukey’s post hoc pairwise comparison was performed for all experimental groups to evaluate the recorded growth factors, degradation rate, enzymatic activities (AChE and GST), and in vitro assays. Each group was analyzed using at least three replicates (five replicates for the in vivo assay). P-values < 0.05 were considered statistically significant. Data were visualized, when applicable, using R software (version 3.2.5).
- [Lines 301–312, Results – Identification]
Figure 2: the legend refers to “neighbor-joining,” while the Methods section specifies UPGMA. Please unify the terminology.
Response: The method used to construct phylogenetic tree was corrected in material section.
- [Lines 394–405, in vivo bacterial count]
The results are described only qualitatively (“significant differences,” “substantial variations”). Please provide the exact values and specify the statistical tests used.
Response: Lines 398-405 (Screening of biodegradation capabilities by chlorpyrifos-degrading isolates to other pesticides), this is a qualitative analysis conducted as a preliminary screening to determine whether our isolates are able to grow on other pesticides. The results were recorded as positive in cases of visible growth and negative in the absence of growth. No statistical analysis was applied, as growth was evaluated qualitatively by the streaking method without numerical measurements. While in case of in vivo bacterial counting, we conducted a quantitative measurements (CFU/g) in lines 481-496.
- [Lines 486–565, Discussion]
The discussion is lengthy and at times repeats the results. Please consider shortening it and placing stronger emphasis on the novelty of the work (isolation of Klebsiella strains and demonstration of their impact on larval survival).
Response: The novelty of the work has been added in discussion Section.
Reviewer 2 Report
Comments and Suggestions for Authors
Materials and Methods
Comment 1 : Sampling : The description of larval collection is clear and precise, including GPS coordinates. However, additional details on the sampling season and environmental conditions (temperature, pesticide history of the field) would strengthen reproducibility and ecological relevance.
Comment 2 : Pesticides : The list of pesticides is adequate, but the suppliers and catalog numbers should be consistently provided. For example, only chlorpyrifos includes the company name (Dow Agro Sciences), while others do not. Consistency is important. The final concentrations (ppm) are mentioned, but the rationale for selecting these concentrations (e.g., based on field doses, or prior studies) should be clarified.
Comment 3: Isolation of chlorpyrifos-degrading gut bacteria : The enrichment culture procedure is generally described, but the details about controls are missing. For example, were uninoculated media controls included to rule out abiotic degradation of chlorpyrifos?
Comment 4 : Screening of biodegradation capabilities by chlorpyrifos-degrading isolates to other pesticides : The design is straightforward, but controls are missing. Were negative controls (uninoculated plates) and positive controls (known pesticide-degrading strains) included? And growth as an indicator of biodegradation may be misleading. Additional confirmation (e.g., pesticide residue measurement by GC–MS) would strengthen the conclusions.
Comment 5: In vivo chlorpyrifos biodegradation assay : More details are needed to ensure reproducibility:
- How many larvae were used per treatment group in the monoassociated experiment?
- Were all experiments repeated independently (biological replicates)?
- Were antibiotics tested for residual effects on larval survival independent of bacterial colonization?
Comment 6 : The feeding protocol (leaf discs dipped for 45 s) may result in variable bacterial inoculation. Quantification of inoculated bacterial load (CFU/leaf) would add robustness.
Results
Comment 7 : In vitro insecticide biodegradability of isolated gut bacteria : The biodegradation results (Table 1) show promising strain-dependent variation. The report of high degradation rates after one day is striking but should be interpreted carefully, as rapid degradation may partly reflect abiotic processes or loss due to adsorption. Were abiotic controls (uninoculated medium with chlorpyrifos) consistently monitored and reported?
Comment 8 : The decline in degradation efficiency at lower pesticide concentrations (strain 64D) is mentioned but not fully explained. This is an important observation and warrants deeper discussion, possibly linked to enzyme kinetics or substrate inhibition.
Comment 9 : Results would be strengthened by performing statistical comparisons across strains.
4. Screening of biodegradation capabilities toward other pesticides : Major comments
Comment 10 : Table 2 uses “Positive/Negative” growth on MM9 + pesticide as the sole carbon source as evidence of biodegradation. Growth alone can reflect utilization of impurities, surfactants, or carryover nutrients. Please confirm biodegradation analytically (e.g., GC–MS/ECD residue quantification over time, recovery-corrected) and/or show stoichiometric CO₂ evolution or specific metabolite/intermediate formation. Without this, the conclusion that lambda-cyhalothrin is biodegraded by 60D is premature.
Comment 11 : Replace binary calls with degradation percentages (± SD) at defined timepoints, and include kinetic parameters (e.g., k_obs, t₁/₂). This will allow comparison across strains and chemistries.
Comment 12 : The tested levels (0.6 ppm chlorantraniliprole, 0.5 ppm carbamate, 5 ppm lambda-cyhalothrin) should be justified (field-relevant doses? solubility limits?). Provide solvent composition (if any) and verify that solvents alone do not support growth.
5. In vivo chlorpyrifos biodegradation assay
Comment 13 : Bacterial counting in larval gut : Specify n (larvae per group per timepoint), whether counts are biological vs. technical replicates, and show means ± SD (or SEM) with exact p-values and the test used. If multiple groups/timepoints are compared, adjust for multiplicity (e.g., Tukey, Benjamini–Hochberg).
Comment 14 : Clarify the washout period after antibiotics before chlorpyrifos exposure and provide data showing the extent of microbiome suppression (e.g., 16S qPCR total load, or culture-independent metrics).
Comment 15 : Larval weight : Table 3 reports means ± SD; please indicate n per group and confirm normality and equal variance assumptions for the t-test. If assumptions fail, use Mann–Whitney.
Comment 16 : Detoxification enzyme activity (AChE, GST). You state AChE activity was “significantly induced” vs. “highest inhibition” (3.5-fold) for strain 60D at 24 h. Induction and inhibition are opposites; please reconcile. If you report percent inhibition relative to control, define the control and the direction clearly.
Comment 17 : Bacterial counting in larval feces : Inferring “strong ability to degrade chlorpyrifos” from higher CFU in feces (e.g., strain 66D >7 log₁₀ CFU/g) is not supported unless you correlate CFU with fecal metabolite profiles or host residue reductions. CFU reflects survival/colonization, not necessarily degradation.
Comment 18 : Cross-reference with in vitro degradation kinetics; if 66D shows the highest fecal CFU but lower day-1 degradation in vitro, discuss potential explanations (colonization advantage, different expression in vivo, substrate access).
Discussion
Comment 19 : The Discussion is comprehensive but tends to repeat results in detail (e.g., mortality reduction percentages, fecal residues, enzyme fold changes). A more concise presentation would improve readability. Consider summarizing results briefly and focusing on interpretation, mechanisms, and implications.
Comment 20 : High chlorpyrifos residues in feces are interpreted as evidence of detoxification, but this may also reflect excretion of unmetabolized pesticide. Consider clarifying this limitation and suggesting targeted metabolite analysis (e.g., 3,5,6-trichloro-2-pyridinol detection) for future studies.
Comment 21 : The interpretation of AChE activity is somewhat confusing. You report “inhibition” but also use wording like “induction” earlier. Ensure terminology is consistent and that enzyme changes are clearly linked to pesticide exposure and/or bacterial effects.
Comment 22 : The Discussion cites relevant studies (e.g., Bacillus cereus, Enterococcus sp., Stenotrophomonas maltophilia). However, comparisons to these works are mostly descriptive. It would be valuable to highlight how your findings extend or contrast with these studies, especially regarding strain-specific differences in Klebsiella.
Comment 23 : The broader ecological or applied significance is not fully developed. For example, what are the implications of finding Klebsiella spp. as degraders in terms of biosafety (many are opportunistic pathogens) or potential for biocontrol applications?
Figures :
For the figures: in Figure 2, the image should be enlarged to ensure readability. Similarly, in Figure 1, the font size should be increased for better legibility, and the overall image quality should be improved.
Author Response
Reviewer 2
Materials and Methods
Comment 1 : Sampling : The description of larval collection is clear and precise, including GPS coordinates. However, additional details on the sampling season and environmental conditions (temperature, pesticide history of the field) would strengthen reproducibility and ecological relevance.
Response: Thank you for this valuable comment. Additional details regarding the sampling season and pesticide application history of the field are included.
Comment 2 : Pesticides: The list of pesticides is adequate, but the suppliers and catalog numbers should be consistently provided. For example, only chlorpyrifos includes the company name (Dow Agro Sciences), while others do not. Consistency is important. The final concentrations (ppm) are mentioned, but the rationale for selecting these concentrations (e.g., based on field doses, or prior studies) should be clarified.
Response: All information has been added consistently in the revised Materials and Methods section.
Comment 3: Isolation of chlorpyrifos-degrading gut bacteria : The enrichment culture procedure is generally described, but the details about controls are missing. For example, were uninoculated media controls included to rule out abiotic degradation of chlorpyrifos?
Response: Regarding the use of controls in the enrichment culture procedure, uninoculated medium controls were not included in this preliminary enrichment step because the aim was to isolate bacterial strains rather than to assess chlorpyrifos degradation. Abiotic degradation would not result in bacterial growth or isolation; therefore, such controls were not relevant at this stage. However, in the subsequent in vitro degradation experiments, uninoculated controls were included to confirm that chlorpyrifos degradation was attributable to the metabolic activity of the isolated strains.
Comment 4 : Screening of biodegradation capabilities by chlorpyrifos-degrading isolates to other pesticides : The design is straightforward, but controls are missing. Were negative controls (uninoculated plates) and positive controls (known pesticide-degrading strains) included? And growth as an indicator of biodegradation may be misleading. Additional confirmation (e.g., pesticide residue measurement by GC–MS) would strengthen the conclusions.
Response: This study represents a preliminary assessment and screening intended to determine whether our strains can grow in the presence of other pesticides. The results will guide future research to explore their potential for degrading additional pesticides and to quantify degradation rates at different pesticide concentrations. A positive control was not included, as no known pesticide-degrading strain was available. However, a negative control (uninoculated plates) was carried out and written inside the text.
Comment 5: In vivo chlorpyrifos biodegradation assay : More details are needed to ensure reproducibility:
- How many larvae were used per treatment group in the monoassociated experiment?
Response: We used five egg groups per treatment, which equates to approximately 2,000 newly hatched larvae on average.
- Were all experiments repeated independently (biological replicates)?
Response: Yes, all experiments were independently repeated as biological replicates.
- Were antibiotics tested for residual effects on larval survival independent of bacterial colonization?
Response: Thank you for your comment. In this study, we did not specifically test the residual effects of antibiotics on larval survival independent of bacterial colonization. Our focus was limited to measuring larval weight and analyzing the bacterial community composition after antibiotics application.
Comment 6 : The feeding protocol (leaf discs dipped for 45 s) may result in variable bacterial inoculation. Quantification of inoculated bacterial load (CFU/leaf) would add robustness.
Response: In this study, four bacterial suspensions were used, each adjusted to the 106 CFU/mL prior to inoculation to ensure comparable exposure levels. All leaf discs were dipped for a fixed duration (45 s) under identical conditions to minimize variation in bacterial load. Because the focus of this experiment was on assessing bacterial survival and passage through the insect gut, quantification was performed based on colony counts from larval feces rather than from the inoculated leaves. This clarification has been added to the revised Methods section (lines 214-219).
Results
Comment 7: In vitro insecticide biodegradability of isolated gut bacteria : The biodegradation results (Table 1) show promising strain-dependent variation. The report of high degradation rates after one day is striking but should be interpreted carefully, as rapid degradation may partly reflect abiotic processes or loss due to adsorption. Were abiotic controls (uninoculated medium with chlorpyrifos) consistently monitored and reported?
Response: Yes, the degradation rates were calculated after subtracting the values obtained from the abiotic controls (uninoculated medium with chlorpyrifos) from those containing inoculated bacteria. Use the following formula to determine the degradation rate.
where A1 represents the insecticide content in the MM9 medium after bacterial inoculation, and A0 represents the insecticide content in the uninoculated control
Comment 8 : The decline in degradation efficiency at lower pesticide concentrations (strain 64D) is mentioned but not fully explained. This is an important observation and warrants deeper discussion, possibly linked to enzyme kinetics or substrate inhibition.
Response: We added a detailed explanation to the discussion section, our results indicated that Klebsiella pneumoniae strain 64D showed the highest degradation rate after 1 day (80.38%), but its efficiency declined to 25% after 5 days. The decline in strain 64D may reflect enzyme kinetic limitations or substrate/product inhibition, where reduced chlorpyrifos availability or accumulation of metabolites suppresses further enzymatic activity and bacterial growth, reflecting the complex interplay between enzyme induction, substrate concentration, and metabolic feedback regulation.
Comment 9: Results would be strengthened by performing statistical comparisons across strains.
Response: Thank you for your valuable comment, we would like to clarify that the statistical comparisons across strains have already been performed and are presented in Table 1 from the beginning. The lowercase letters (e.g., a, b, c) next to the mean values within each column represent the results of a post-hoc test. As noted in the table's footnote, it means within the same point that do not share a letter are significantly different from one another.
- Screening of biodegradation capabilities toward other pesticides : Major comments
Comment 10 : Table 2 uses “Positive/Negative” growth on MM9 + pesticide as the sole carbon source as evidence of biodegradation. Growth alone can reflect utilization of impurities, surfactants, or carryover nutrients. Please confirm biodegradation analytically (e.g., GC–MS/ECD residue quantification over time, recovery-corrected) and/or show stoichiometric CO₂ evolution or specific metabolite/intermediate formation. Without this, the conclusion that lambda-cyhalothrin is biodegraded by 60D is premature.
Response: We fully agree that analytical confirmation is necessary to substantiate true biodegradation. The current experiment was intended as a preliminary qualitative screening to identify isolates capable of growing in the presence of various pesticides, thereby indicating potential metabolic capacity. Future investigations will focus on quantitative validation of pesticide degradation using analytical techniques (e.g., GC–MS or HPLC) and metabolic assays to confirm and characterize the underlying degradation pathways.
Comment 11 : Replace binary calls with degradation percentages (± SD) at defined timepoints, and include kinetic parameters (e.g., k_obs, t₁/₂). This will allow comparison across strains and chemistries.
Response: Thank you for your valuable comment. The aim of our study was to calculate the degradation rate of each bacterial isolate, using the same method described by Li et al. (2020). The inclusion of kinetic parameters (e.g., kₒᵦₛ and t₁/₂) will be considered in future studies.
Li H, Qiu Y, Yao T, Ma Y, Zhang H, Yang X, Li C: Evaluation of seven chemical pesticides by mixed microbial culture (PCS-1): Degradation ability, microbial community, and Medicago sativa phytotoxicity. Journal of hazardous materials 2020, 389:121834.
Comment 12 : The tested levels (0.6 ppm chlorantraniliprole, 0.5 ppm carbamate, 5 ppm lambda-cyhalothrin) should be justified (field-relevant doses? solubility limits?). Provide solvent composition (if any) and verify that solvents alone do not support growth.
Response: Thank you for your comment. We did not use any solvent in the experiments; all insecticides were dissolved directly in distilled water. The tested concentrations (0.6 ppm chlorantraniliprole, 0.5 ppm carbamate, and 5 ppm lambda-cyhalothrin) correspond to the field-recommended doses.
- In vivo chlorpyrifos biodegradation assay
Comment 13 : Bacterial counting in larval gut : Specify n (larvae per group per timepoint), whether counts are biological vs. technical replicates, and show means ± SD (or SEM) with exact p-values and the test used. If multiple groups/timepoints are compared, adjust for multiplicity (e.g., Tukey, Benjamini–Hochberg).
Response: Bacterial counting was performed using 10 larvae per group, with a total of 30 larvae analyzed. Each larva represented a biological replicate. Data are presented as mean ± SD (log₁₀ CFU/mL) in Figure 6. Statistical analysis was conducted using one-way ANOVA followed by Tukey’s post hoc pairwise comparison, as described in the Materials and Methods section (Data analysis). Significant differences were considered at p < 0.01, as indicated in Figure 6.
Comment 14 : Clarify the washout period after antibiotics before chlorpyrifos exposure and provide data showing the extent of microbiome suppression (e.g., 16S qPCR total load, or culture-independent metrics).
Response: Thank you for this valuable comment. The washout period between antibiotic treatment and chlorpyrifos exposure was three days. To validate the effect of antibiotics on the gut microbiota, gut bacterial counts were determined after antibiotic treatment and prior to chlorpyrifos exposure, confirming a noticeable reduction in bacterial load.
Comment 15 : Larval weight : Table 3 reports means ± SD; please indicate n per group and confirm normality and equal variance assumptions for the t-test. If assumptions fail, use Mann–Whitney.
Response: Thank you for your valuable comment. We used 30 larvae in each group (n = 30). Additionally, We confirm that the assumptions for the t-test were met; the data were checked and found to be normally distributed (Shapiro-Wilk test) and possessed equal variances (Levene's test). Therefore, the independent samples t-test was the appropriate statistical method for this analysis.
Comment 16 : Detoxification enzyme activity (AChE, GST). You state AChE activity was “significantly induced” vs. “highest inhibition” (3.5-fold) for strain 60D at 24 h. Induction and inhibition are opposites; please reconcile. If you report percent inhibition relative to control, define the control and the direction clearly.
Response: Thank you for your careful observation. It has been corrected and clarified in the revised manuscript.
Comment 17 : Bacterial counting in larval feces : Inferring “strong ability to degrade chlorpyrifos” from higher CFU in feces (e.g., strain 66D >7 log₁₀ CFU/g) is not supported unless you correlate CFU with fecal metabolite profiles or host residue reductions. CFU reflects survival/colonization, not necessarily degradation.
Response: Day 1: No meaningful linear relationship exists between residue concentration and CFU count. The data appear widely scattered.
Day 3: A significant inverse correlation indicates that as residue levels increase, CFU tends to decrease markedly suggesting anti-activity effects may become pronounced after 3 days.
Day 5: A weak, non-significant positive trend appears, potentially reflecting recovery or adaptation of effect .
Comment 18 : Cross-reference with in vitro degradation kinetics; if 66D shows the highest fecal CFU but lower day-1 degradation in vitro, discuss potential explanations (colonization advantage, different expression in vivo, substrate access).
Response: Interestingly, K. pneumoniae strain 66D exhibited the highest fecal CFU in vivo, while showing a comparatively lower Day-1 in vitro degradation rate (51.3 ± 0.15 %) relative to strains 60D (77.9 ± 0.91 %) and 64D (80.38 ± 0.68 %). This apparent discrepancy suggests that high colonization potential does not necessarily correspond to greater degradation efficiency under defined in vitro conditions. Such differences likely arise from the distinct selective pressures and environmental contexts of the host gut compared to laboratory assays. In vivo colonization success may depend on factors such as adhesion ability, biofilm formation, immune evasion, and resource utilization rather than metabolic rate alone. Moreover, gene expression for degradative enzymes may be differentially regulated in response to host cues, nutrient gradients, or microbiota-derived metabolites, which are absent in vitro. Additionally, substrate access and compound bioavailability differ markedly between homogeneous in vitro media and the complex intestinal matrix. Therefore, the high in vivo CFU of strain 66D may reflect a colonization advantage or enhanced in vivo expression of degradation-related genes that are not induced in vitro. Future investigations combining transcriptional profiling (RNA-seq/qPCR) and metabolite quantification in colonized larvae are warranted to distinguish between these potential mechanisms.
Discussion
Comment 19 : The Discussion is comprehensive but tends to repeat results in detail (e.g., mortality reduction percentages, fecal residues, enzyme fold changes). A more concise presentation would improve readability. Consider summarizing results briefly and focusing on interpretation, mechanisms, and implications.
Response: Thanks for your insightful comment. Some explanations have been added to the discussion section.
Comment 20 : High chlorpyrifos residues in feces are interpreted as evidence of detoxification, but this may also reflect excretion of unmetabolized pesticide. Consider clarifying this limitation and suggesting targeted metabolite analysis (e.g., 3,5,6-trichloro-2-pyridinol detection) for future studies.
Response: Thank you for this valuable comment. Your recommendation will be considered in future studies.
Comment 21 : The interpretation of AChE activity is somewhat confusing. You report “inhibition” but also use wording like “induction” earlier. Ensure terminology is consistent and that enzyme changes are clearly linked to pesticide exposure and/or bacterial effects.
Response: This error has been corrected in the results section, and the terminology has been checked and standardized throughout the manuscript.
Comment 22 : The Discussion cites relevant studies (e.g., Bacillus cereus, Enterococcus sp., Stenotrophomonas maltophilia). However, comparisons to these works are mostly descriptive. It would be valuable to highlight how your findings extend or contrast with these studies, especially regarding strain-specific differences in Klebsiella.
Response: The role of Proteobacteria in chlropyrifos resistance has been added to the Discussion section; however, no specific studies have investigated Klebsiella spp. isolated from the insect gut.
Comment 23 : The broader ecological or applied significance is not fully developed. For example, what are the implications of finding Klebsiella spp. as degraders in terms of biosafety (many are opportunistic pathogens) or potential for biocontrol applications?
Response: The identification of Klebsiella spp. as pesticide degraders highlights both ecological and biosafety considerations. While their metabolic versatility supports potential use in biodegradation, their opportunistic pathogenicity limits direct application. Future work should focus on exploiting their degradation genes in non-pathogenic hosts or cell-free enzymatic systems to ensure safe bioremediation approaches.
Figures :
For the figures: in Figure 2, the image should be enlarged to ensure readability. Similarly, in Figure 1, the font size should be increased for better legibility, and the overall image quality should be improved.
Response: The figures 1 and 2 were enlarged and attached inside the manuscript.
Reviewer 3 Report
Comments and Suggestions for Authors
The manuscript “Isolation and Characterization of Chlorpyrifos-Degrading Gut Bacteria from Field-
Collected Larvae of Spodoptera frugiperda (J.E. Smith) (Lepidoptera: Noctuidae)” describes the
evaluation of the chlorpyrifos-degrading capacity of four bacterial strains isolated from the gut
of Spodoptera frugiperda larvae, both in vitro and in vivo. This study addresses a highly relevant topic:
the biodegradation of a widely used pesticide by using natural biological processes, such as the
microbiota of pesticide-resistant organisms. Also, the study assesses the role and mechanism of
action of the evaluated strains in chlorpyrifos degradation, and the influence of factors such as
temperature, pH, and incubation time on the survival of the studied strains.
Although the biodegradation of this pesticide using gut-isolated microorganisms from pest larvae has
been previously reported, this work uses a different Spodoptera species and isolates bacterial taxa
distinct from those described in previous studies, which provides novelty and contributes to the
understanding of pest resistance mechanisms. Appropriate statistical analyses were conducted and
are clearly presented, and the article is suitable for publication. However, some improvements need
to be made:
• In line 25 (“In this study, we successfully isolated four isolates of chlorpyrifos-degrading
bacteria…”) sounds redundant; the sentence should be rewritten.
• In lines 34–38: The sentence “To address this principle i) isolation and identification of the
chlorpyrifos-degrading gut bacteria from the field collection of the invasive pest S. frugiperda
larvae. ii) Evaluation of chlorpyrifos biodegradation ability through in vitro assay. iii) Assess
the impact of particular bacterial taxa that have the ability to degrade chlorpyrifos directly in
the gut” could be changed for clarity as follows: “Three strategies were implemented to
address this principle: (i) isolation and identification of chlorpyrifos-degrading gut bacteria
from field-collected S. frugiperda larvae; (ii) evaluation of chlorpyrifos biodegradation
capacity through in vitro assays; and (iii) assessment of the impact of specific bacterial taxa
capable of degrading chlorpyrifos directly within the gut.”
• Section 5.2 (Larval weight ) is very short. The effects of the antibiotics could be discussed
more extensively.
• Although the references are appropriate, the format of the cited websites needs to be revised.
• The tables and figures are helpful to understand the data; however, Figures 1, 2, and 9 have
small letters that look blurry and may affect readability.
• In the Materials and Reagents section, the notation for E.C. should be standardized.
• Lines 108 and 203: Alcohol should be referred to as a disinfectant rather than a sterilizer,
since it does not eliminate all types of microorganisms.
• Line 249 (“AChE activity was determined following the method of …”) sounds like it was
incomplete and should be clarified.
• Although the conclusions were supported by the results, the identified detoxification
mechanism (AChE inhibition) should be mentioned briefly.
• Section 5.2 (Larval weight ) is too short. The effects of the antibiotics effects could be
discussed more extensively
Author Response
Reviewer 3
The manuscript “Isolation and Characterization of Chlorpyrifos-Degrading Gut Bacteria from Field- Collected Larvae of Spodoptera frugiperda (J.E. Smith) (Lepidoptera: Noctuidae)” describes the evaluation of the chlorpyrifos-degrading capacity of four bacterial strains isolated from the gut of Spodoptera frugiperda larvae, both in vitro and in vivo. This study addresses a highly relevant topic: the biodegradation of a widely used pesticide by using natural biological processes, such as the microbiota of pesticide-resistant organisms. Also, the study assesses the role and mechanism of action of the evaluated strains in chlorpyrifos degradation, and the influence of factors such as temperature, pH, and incubation time on the survival of the studied strains.
Although the biodegradation of this pesticide using gut-isolated microorganisms from pest larvae has been previously reported, this work uses a different Spodoptera species and isolates bacterial taxa distinct from those described in previous studies, which provides novelty and contributes to the understanding of pest resistance mechanisms. Appropriate statistical analyses were conducted and are clearly presented, and the article is suitable for publication. However, some improvements need to be made:
In line 25 (“In this study, we successfully isolated four isolates of chlorpyrifos-degrading
bacteria…”) sounds redundant; the sentence should be rewritten.
Response: Thank you for your observation. The sentence has been revised.
In lines 34–38: The sentence “To address this principle i) isolation and identification of the
chlorpyrifos-degrading gut bacteria from the field collection of the invasive pest S. frugiperda
larvae. ii) Evaluation of chlorpyrifos biodegradation ability through in vitro assay. iii) Assess
the impact of particular bacterial taxa that have the ability to degrade chlorpyrifos directly in
the gut” could be changed for clarity as follows: “Three strategies were implemented to
address this principle: (i) isolation and identification of chlorpyrifos-degrading gut bacteria
from field-collected S. frugiperda larvae; (ii) evaluation of chlorpyrifos biodegradation
capacity through in vitro assays; and (iii) assessment of the impact of specific bacterial taxa
capable of degrading chlorpyrifos directly within the gut.”
Response: Thank you for your helpful suggestion. The sentence has been revised as recommended.
Section 5.2 (Larval weight ) is very short. The effects of the antibiotics could be discussed
more extensively.
Response: Discussed in lines 431-434 in results section.
Although the references are appropriate, the format of the cited websites needs to be revised.
Response: Corrected to be https://blast.ncbi.nlm.nih.gov/Blast.cgi (accessed on 28 March 2024). in line 138 in Materials and Methods section.
The tables and figures are helpful to understand the data; however, Figures 1, 2, and 9 have
small letters that look blurry and may affect readability.
Response Figures 1, 2 and 9 have been enlarged and replaced with higher-resolution versions in the revised manuscript.
In the Materials and Reagents section, the notation for E.C. should be standardized.
Response: Thank you for your comment. The notation for EC (emulsifiable concentrate) has been standardized.
Lines 108 and 203: Alcohol should be referred to as a disinfectant rather than a sterilizer,
since it does not eliminate all types of microorganisms.
Response: Thank you for your observation. The term has been corrected to “disinfectant” in the revised manuscript as suggested.
Line 249 (“AChE activity was determined following the method of …”) sounds like it was
incomplete and should be clarified.
Response: The author's name was added to complete the sentence.
Although the conclusions were supported by the results, the identified detoxification
mechanism (AChE inhibition) should be mentioned briefly.
Response: Thank you for your comment, it has been briefly mentioned in the conclusions section.
Section 5.2 (Larval weight) is too short. The effects of the antibiotics effects could be
discussed more extensively
Response: Discussed in lines 431-434 in results section.